# Methods for Weighting Decisions to Assist Modelers and Decision Analysts: A Review of Ratio Assignment and Approximate Techniques

Barry Ezell [1,*], Christopher J. Lynch [1] and Patrick T. Hester [2]

1   Virginia Modeling, Analysis and Simulation Center, Old Dominion University, Suffolk, VA 23435, USA; cjlynch@odu.edu
2   Modus Operandi Inc., Melbourne, FL 32901, USA; phester@modusoperandi.com
*   Correspondence: bezell@odu.edu

**Abstract:** Computational models and simulations often involve representations of decision-making processes. Numerous methods exist for representing decision-making at varied resolution levels based on the objectives of the simulation and the desired level of fidelity for validation. Decision making relies on the type of decision and the criteria that is appropriate for making the decision; therefore, decision makers can reach unique decisions that meet their own needs given the same information. Accounting for personalized weighting scales can help to reflect a more realistic state for a modeled system. To this end, this article reviews and summarizes eight multi-criteria decision analysis (MCDA) techniques that serve as options for reaching unique decisions based on personally and individually ranked criteria. These techniques are organized into a taxonomy of ratio assignment and approximate techniques, and the strengths and limitations of each are explored. We compare these techniques potential uses across the Agent-Based Modeling (ABM), System Dynamics (SD), and Discrete Event Simulation (DES) modeling paradigms to inform current researchers, students, and practitioners on the state-of-the-art and to enable new researchers to utilize methods for modeling multi-criteria decisions.

**Keywords:** multi-criteria decision making; multi-criteria decision analysis; attribute weighting; MCDM; MCDA

## 1. Introduction

Decision-making processes exist in many forms within computational models and simulations, and varied needs drive the scope of how decision-making is represented across modeling paradigms. Models are commonly developed under a specified context or experimental frame [1,2], verified, validated, and tested under that perspective [3,4], and decision-making representations' forms are dependent upon the utilized perspective. The desired levels of realism and aggregation for a model influences its form and impacts the insights that can be gleaned from a model when simulated. Depending on the modeled system, decisions can be responsible for dynamically altering the structure of the simulated environment, modifying behaviors or goals of simulated entities, ascertaining group membership selections, and determining the results of actions or interactions.

Many factors can lead to differences in the desired type of decision-making representation utilized within a model, including: differences in the identified relevant model context [5–7]; differences in stakeholder perspectives [8–10]; differing perspectives on the importance of rare events versus likely outcomes [11]. Keeney and Raiffa [12] describe decision analysis as a "prescriptive approach...to think hard and systematically about some important real problems" [12]. Thus, at its core, it helps us to understand how we should make decisions. It is the formal process of choosing from among a candidate set of decisions to determine which alternative is most valuable to the decision maker. Complicating most real decisions is that we wish to achieve multiple aims; that is, we evaluate

candidate solutions based on a potentially large number of criteria. For an apartment, we may consider cost, size, location, and amenities, whereas a job selection problem may cause us to consider salary, growth potential, location, and benefits. Each of these criteria is important to us and we need to consider them comprehensively to evaluate our problem using multi-criteria decision analysis (MCDA).

MCDA is already being applied to assist in modeling and simulating systems involving decisions made by individuals or groups. MCDA has been widely applied for natural resource management as it provides a structured approach for integrating key management factors, captures the multi-functional uses of forests, and accounts for multiple stakeholder perspectives on how to best manage the forest [13]. With respect to public health safety, MCDA has been utilized to explore preventive programs for the prevention of Lyme disease [14] and to assess programs for preventing the spread of West Nile virus [15]. Scholten, Maurer [16] compares the use of MCDA models against integrated assessment models in identifying alternatives for long term water supply planning at a town scale. Their study finds that all the models identified provided better performance than the current water supply system; however, the MCDA models also provided better value ranges and formed better bases for discussion than the integrated assessment models.

We review and present a representative sample of commonplace techniques within MCDA. Our selection process considers techniques that are common for situations where there exist only a few attributes as well as situations where there may be many attributes. Situations pertaining to the representation of only a few attributes lend themselves to ratio assignment techniques, while scenarios with many attributes lend themselves to approximate techniques. The techniques reviewed and presented in our taxonomy rely on expert judgement, have been used in practice for over 30 years, and have been utilized in the personal experience of the authors in many research projects [17–20]. We discuss how these MCDA techniques can be utilized for modeling decision making within three common modeling paradigms. Our objective is to improve understanding of how approximate and ratio assignment techniques can be used to expand the existing decision modeling toolboxes within these modeling paradigms, as well as the circumstances under which the techniques are applicable.

Modeling paradigms represent decision making in a variety of ways, capture decisions at different levels of granularity, and generate different responses with respect to how a decisions' outcome impacts a simulation. For instance, System Dynamics (SD) represent decisions based on nonlinear population behaviors aggregated as flow rates over time [21]. Decision making is reflected at the system level through information feedback and delays [22]. Discrete Event Simulation (DES) captures decisions at the system design level [23,24]. The DES decision processes represent the aggregated options for how entities traverse within the modeled system [25,26]. Agent Based Models (ABM) represent decision-making at the individual level [27], with agents' decisions based on their goals and their current states. Aggregate system behavior is examined based on how the collective interactions lead to system level behaviors over time [28,29]. Methods for representing decisions include the use of rules [30,31], knowledge architectures [32], state charts [33,34], temporal belief logic [35], decision nodes [25], and decision trees [36], to name a few options. Time independent paradigms such as Markov chains, Bayesian inference, Petri Nets, and Hidden Markov Models can provide instantaneous decision selections based on the current state of known information without required time dependencies [37–39]. Additionally, model stakeholders and model builders can arrive at different validity constraints based on the model context combined with their own experiences [6]. This can lead to different preferences for how decision-making should be specified within a simulation.

Many techniques have been established for modeling decisions and selecting the appropriate technique should involve examining how the decision is made within the real system [21,36,40–43]. This involves examining how decisions are made and knowing what the set of possible decision options includes. The literature on individual techniques and MCDA in general is vast. Multi-criteria decision models have been applied to study

fall protection support for construction sites [44], temperature-aware routing in wireless body area networks [45], performance assessment of credit granting decision systems [46], assessment of player rankings in E-Sports [47], load profiling for power systems [48], identification of ideal business location selection [49], performance of emergency systems under COVID-19, [50] venture investment [51], failure modes analysis [52], group decision making [53], drug trafficking [54], and remote sensing for drought characterization [55]. This article focuses on summarizing common weighting methods, including their advantages and disadvantages, to aid the reader in the determination of an appropriate method for use given the particulars of a given decision to be made. This summary is extended to discuss the applicability of MCDA techniques for use in decision making within a sample of commonly utilized modeling paradigms. Research and practical application have shown that additive models are the most extensively used model in multi-criteria decision analysis [56]. However, a review of these techniques uses and applications within M&S has not been conducted. We provide an assessment of the state-of-the-art of MCDA weighting methods, as well as a comparison analysis of the use of these methods in the context of a realistic problem. Throughout this article, any person, such as model builders, model stakeholders, risk managers, engineers, and decision makers, eliciting attribute weights is referred to as user.

## 2. Materials and Methods

MCDA assumes preferential independence among criteria and many weighting methods are built on the assumption of the use of an additive value function. Identifying appropriate means of calculating the weightings of criteria involved in a decision, such as sampling from a uniform distribution [57] or the use of rankings [58], is important for differentiating the category of MCDA technique that is suitable for the simulation. As such, we conduct our assessment under the assumption that an additive preference model of the form provided in Equation (1) is being utilized to inform decision making.

$$v(x) = \sum_{i=1}^{n} w_i v_i(x_i) \tag{1}$$

where $n$ is the total number of criteria being considered, $w_i$ is the weight of the $i$-th criteria, $v_i(x_i)$ is the $i$-th value function, and $x$ is the vector of all criteria values. Within a value function, all weights must add up to 1:

$$\sum_{i=1}^{n} w_i = 1 \tag{2}$$

The use of an additive utility model requires that criteria used in the model are mutually preferentially independent [59]. This means that the weight assigned to any particular criterion is not influenced by the weight assigned to any other attributes. For example, consider choosing between departure times of 6 and 10 a.m. for a flight and their respective costs are $250 and 300. Mutually preferentially independent means that you prefer the cheaper flight to the more expensive one regardless of departure time, and you prefer the later flight to the earlier one regardless of cost. If you prefer the later flight regardless of the ticket price, however, the price dictates your departure preference, then departure time is preferentially independent of cost, but they are not mutually preferentially independent. If the attributes are not mutually preferential independent, there are techniques to combine them using a joint utility function that is beyond the scope of this paper. Additionally, all criteria should be mutually exclusive and collectively exhaustive, that is, they represent the entirety of relevant criteria and each is independent of all others.

A taxonomy has been developed to categorize MCDA techniques to help in identifying and conveying the similarities and differences in uses, benefits, and limitations of each of the techniques. We concentrate on two general approaches for assigning weights to preference attributes within the context of a multi attribute utility model: ratio assignment and approximate techniques. Figure 1 provides the two primary classifications of techniques within MCDA, with each followed by four technique classifications.

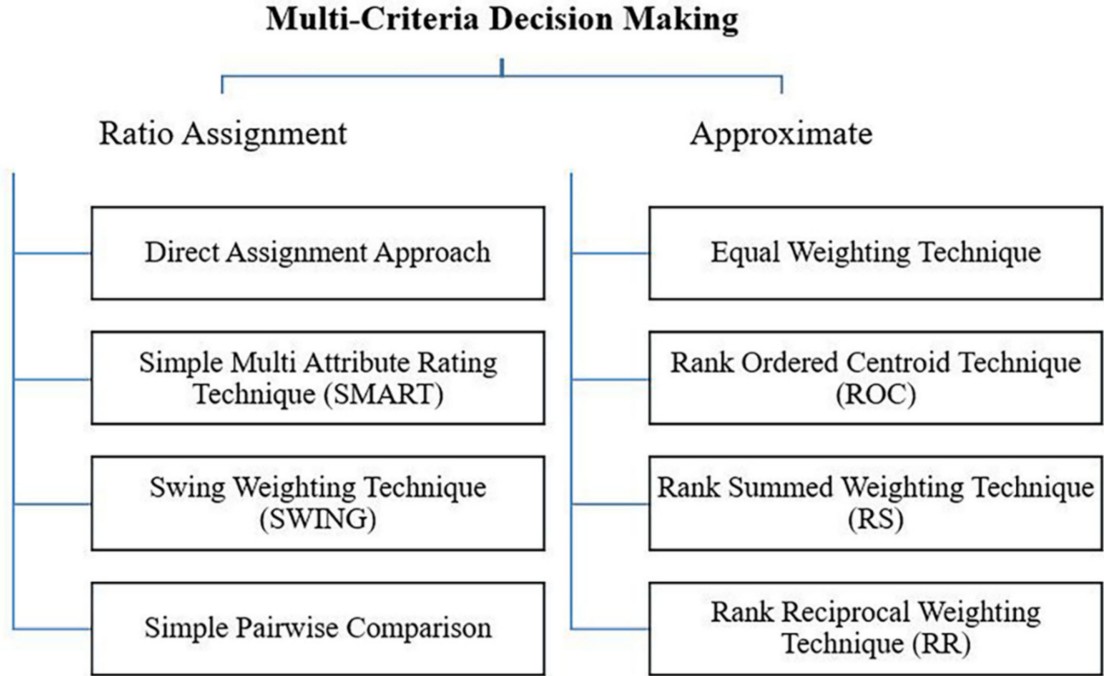

**Figure 1.** Taxonomy of Multi-Criteria Decision Analysis techniques.

Within each branch of the taxonomy, we have selected the techniques which are most usable in practice. Note that some techniques may be better but are unwieldy. Within each category, the techniques are ordered first by direct and then indirect methods, and then notionally from easiest for the decision maker to implement and use to the more difficult weighting methods, requiring more time and resources to set up the weights for the attributes. Borcherding, Eppel [60] suggest that the ratio, swing, tradeoff, and pricing out methods are most commonly used in practice for MCDA. However, more recently, researchers have focused on direct methods for determining weights, including equal and rank-order weighting methods. Weights are often obtained judgmentally with indirect methods [61]; therefore, direct methods that remove some of the subjectivity while determining appropriate weights have become increasingly popular.

The difference between ratio assignment and approximate techniques lies in the nature of the questions posed to elicit weights. Ratio assignment techniques assign a score to each attribute based on its absolute importance relative to a standard reference point or relative importance with respect to other attributes. The resulting weights are obtained by taking the ratio of each individual attribute score to the sum of the scores across all attributes. Approximate techniques assign an approximate weight to each attribute, strictly according to their ranking relative to other attributes with respect to importance. Approximate techniques appeal to principles of order statistics to justify weights in the absence of additional information on relative preference.

As observed in the literature over the past three decades, there are several important pitfalls to be aware of when assigning attribute weights as described below:

- Objective and attribute structure. The structure of the objectives and the selection of weighting methods affect results and should be aligned to avoid bias;
- Attribute definitions affect weighting. The detail with which certain attributes are specified affects the weight assigned to them; that is, the division of an attribute can increase or decrease the weight of an attribute. For example, weighing price, service level, and distance separately as criteria for a mechanic selection led to different results than weighing shop characteristics (comprised of price and service level) and distance did [62];

- Number of attributes affects method choice. It is very difficult to directly or indirectly weight when one has to consider many attributes (e.g., double digits or more), owing to the greater difficulty associated with answering all the questions needed for developing attribute weights; Miller [63] advocates the use of five to nine attributes to avoid cognitive overburden;
- More attributes are not necessarily better. As the number of attributes increases, there is a tendency for the weights to equalize, meaning that it becomes harder to distinguish the difference between attributes in terms of importance as the number of significant attributes increases [64];
- Attribute dominance. If one attribute is weighted heavier than all other attributes combined, the correlation between the individual attribute score and the total preference score approaches one;
- Weights compared within but not among decision frameworks. The interpretation of an attribute weight within a particular modeling framework should be the same regardless of the method used to obtain weights [65]; however, the same consistency in attribute weighting cannot be said to be present across all multi-criteria decision analysis frameworks [66];
- Consider the ranges of attributes. People tend to neglect accounting for attribute ranges when assigning weights using weighting methods that do not stress them [56,67]; rather, these individuals seem to apply some intuitive interpretation of weights as a very generic degree of importance of attributes, as opposed to explicitly stating ranges, which is preferred [68–70]. This problem could occur when evaluating job opportunities. People may assume that salary is the most important factor, however, if the salary range is very narrow (e.g., a few hundred dollars), then other factors such as vacation days or available benefits may in fact be more important in the decision maker's happiness.

There is no superior method for eliciting attribute weights, independent of a problem's context. Consequently, users should be aware of how each method works, its drawbacks and advantages, the types of questions asked by the method, how these answers are used to generate weights, and how different the weights might be if other methods are used. Peer reviewers should be mindful of how each of these methods for eliciting attribute weights are used in practice and how users of these methods interpret the results. The specific method for eliciting attribute weights itself is not the only ingredient in stimulating the discussion. The weighting methods are only tools used in the analysis, and one should focus on the process for how the weights are used [65].

Using the defined taxonomy from Figure 1, we next evaluate the characteristics of each of the specified MCDA classifications. The strengths and weaknesses of each categorization are explored and the criteria for using the techniques are presented. Examples of use are discussed in order to convey the context under which the techniques are applicable. This is followed by a discussion of each techniques' potential uses within the purview of computational modeling within the paradigms of ABM, DES, and SD.

## 3. Results

### 3.1. Ratio Assignment Techniques

Ratio Assignment Techniques ask decision makers questions where answers imply a set of weights corresponding to the user's subjective preferences. The result of this questioning is a set of scores, or points, assigned to each attribute from which the corresponding weights are calculated after normalizing each attribute score with respect to the total score across all attributes. A variety of ratio assignment techniques exist, including: (1) direct assignment technique (DAT), (2) simple multi attribute rating technique (SMART) and its variants, (3) swing weight technique (SWING), and (4) simple pairwise comparison (PW). Each is discussed in the following subsections. The method is introduced, its steps are described, and then its strengths and limitations are presented.

### 3.1.1. Direct Assignment Technique (DAT)

The Direct Assignment Technique (DAT) asks users to assign weights or scores directly to preference attributes. For example, the user may need to divide a fixed pot of points (e.g., 100) among the attributes. Alternatively, users may also be asked to score each attribute over some finite scale (e.g., 0 to 100) and the resulting weights are then calculated by taking the ratio of individual scores to the total score among all attributes.

The Direct Assignment Technique is comprised of the following two steps: (1) assign points to each attribute, and (2) normalize the points such that the total is equal to one.

### DAT Step 1: Assign Points to Each Attribute

One of two approaches can be adopted for completing this step. The first approach considers a fixed pot of points and asks users to divide the pot among the attributes where attributes of greater importance receive higher scores than those of lesser importance. For example, if the total pot consists of 100 points, users would assign a portion of this total among the set of attributes.

The second approach considers a finite range of potential scores and asks user to assign a point value to each attribute according to its importance, where higher importance attributes receive more points than those of lesser importance. For example, if a range of scores ranging from 0 to 100 is considered, users would choose a point value between these limits to establish the relative importance among attributes.

As previously mentioned, it is important that an objective is established and that the ranges (swing) for each attribute are defined; therefore, a common example will be used throughout this paper. We will consider the purchase of a car for a small family, early in their careers, with one small child and a short commute. One could see that the relative weighting of attributes might change if the problem definition changed, e.g., if the decision maker had a long commute or large family.

The attributes will use notional ranges for the remainder of this paper. Again, the relative weight that a decision maker would apply may be impacted by the range. A narrow purchase price range of $20,000 to $20,200 would have less importance than that of a larger range. The criteria used for the analysis of this choice are Purchase Price, Attractiveness, Reliability, Gas Mileage, and Safety Rating. Assume a fixed pot of 1000 points to be divided among the five attributes. These criteria, their abbreviations, their least and most preferred value, and scores are shown in Table 1.

**Table 1.** Criteria for decision problem.

| Abbreviation | Criteria | Least Preferred | Most Preferred | Score |
|:---:|:---:|:---:|:---:|:---:|
| (P) | Purchase Price | $30,000 | $15,000 | 400 points |
| (R) | Reliability (Initial Owner complaints) | 150 | 10 | 300 points |
| (S) | Safety | 3 star | 5 star | 150 points |
| (A) | Attractiveness (qualitative) | Low | High | 100 points |
| (G) | Gas Mileage | 20 mpg | 30 mpg | 50 points |
| (P) | Purchase Price | $30,000 | $15,000 | 400 points |

### DAT Step 2: Calculate Weights

Using the point scores assigned to each of the attributes in the previous step, the second step of the Direct Assignment Technique is to calculate attribute weights. This is done by normalizing each attribute score against the total score among all attributes as shown in Equation (3).

$$w_i = S_i / \sum_j S_j \tag{3}$$

In the car buying example, based on a fixed budget of points, the weights for each attribute can be readily calculated using Equation (3). This result is shown in Table 2.

**Table 2.** DAT weight evaluation.

| Abbreviation | Criteria | Formula | Weight |
|:---:|:---:|:---:|:---:|
| (P) | Purchase Price | 400/1000 | =0.40 |
| (R) | Reliability | 300/1000 | =0.30 |
| (S) | Safety | 150/1000 | =0.15 |
| (A) | Attractiveness | 100/1000 | =0.10 |
| (G) | Gas Mileage | 50/1000 | =0.05 |
| | Sum | 1000 points | =1.00 |

Strengths of This Approach

This approach is the most straightforward of the techniques presented in this paper for eliciting attribute weights in that it does not require the user to formally establish a rank order of attributes a priori. The number of questions needed to assign weights using the direct assignment technique is equal to the number of preference attributes. Thus, the effort required to obtain attribute weights scales linearly with the number of attributes.

Limitations of This Approach

Weights determined using the first approach (divide the pot) must be recalculated if new attributes are added or old ones are removed. However, this limitation does not apply to the second approach (allocation of absolute points). The second approach for assigning weights (allocation of absolute points) is performed without reference to any particular reference point. Yet, establishing a reference point is something humans need to do in order to make quantitative comparisons. For example, without a specified reference point, the user may use his or her best judgment to define what a particular score means (e.g., 0, 50, or 100 on a 100-point scale) and use one or more of these as a basis for assigning scores to attributes. Unfortunately, this approach is sensitive to the chosen reference point and assigned definition, and may produce weights that differ widely between users. One approach to alleviate this limitation is to establish a well-defined constructed scale showing what different scoring levels mean.

3.1.2. Simple Multi Attribute Rating Technique (SMART)

The Simple Multi Attribute Rating Technique (SMART) [71,72] is an approach for determining weighting factors indirectly through systematic comparison of attributes against the one deemed to be the least important. SMART consists of two general activities: (1) rank order attributes according to the relative importance overall, and (2) select either the least or most important attribute as a reference point and assess how much more or less important the other attributes are with respect to the reference point. This step involves calculating attribute weights from ratios of individual attribute scores to the total score across all attributes.

Methodological improvements to SMART, known as SMARTS and SMARTER, were proposed by Edwards and Barron (73). SMARTS (SMART using Swings) uses linear approximations to single-dimension utility functions, an additive utility model, and swing weights to improve weighting [73]. SMARTER (SMART Exploiting Ranks) builds on SMARTS but substitutes the second of the SMARTS swing weighting steps, instead using calculations based on ranks.

The SMART technique is comprised of the following four steps: (1) rank order attributes, (2) establish the reference attribute, (3) estimate the importance of other attributes with respect to the reference attribute, and (4) calculate weights.

SMART Step 1: Rank Order Attributes

Consider a finite set of attributes or criteria deemed relevant by an individual or group of experts to a particular decision problem. This first step asks experts to agree on a rank ordering of these attributes according to their relative contribution to the expert's overall preference within an additive utility (or value) function framework. Ranking can be either

from most to least important or from least to most important. A number of approaches exist to assist in holistic ranking, the most popular and well known being pairwise ranking [74].

For example, consider our automobile purchase problem. The output from Elicitation Step 1 would be a rank ordering of the five relevant criteria from least to most important as shown in Table 3.

**Table 3.** Rank ordering of decision criteria.

| Abbreviation | Criteria | Formula |
|:---:|:---:|:---:|
| (G) | Gas Mileage | 1 |
| (A) | Attractiveness | 2 |
| (S) | Safety | 3 |
| (R) | Reliability | 4 |
| (P) | Purchase Price | 5 |

SMART Step 2: Establish the Reference Attribute

In this second step, experts select a common reference attribute, assign it a fixed score, and estimate the extent to which the remaining attributes are more or less important than the reference attribute. Any attribute can assume the role of reference attribute. For SMART, however, it is common to assign this role to the least important attribute and assign a reference score of 10 points.

SMART Step 3: Score Attributes Relative to the Reference Attribute

Given a fixed reference attribute (i.e., the least important attribute), experts are asked how much more important the remaining attributes are with respect to the reference attribute. For example, if the least important attribute is used as the reference point with a reference score of 50 points, experts would be asked to judge how many points should be allocated to each remaining attribute with respect to this reference attribute in a relative sense (e.g., 50 more points) or absolute sense (e.g., 100 points). It is common to systematically evaluate the remaining attributes in order of increasing importance to ensure individual or group consistency between the results from this step and the ordinal rankings from step 1; however, it may be worthwhile to randomize the order in which attributes are assessed as a means for uncovering any inconsistencies in preference.

Consider the car buying example discussed in Step 1. Using the least important attribute (i.e., Gas Mileage) as the reference attribute with a reference score of 50 points, point scores could be assigned to the remaining attributes as shown in Table 4.

**Table 4.** Total points for decision criteria.

| Abbreviation | Criteria | Points | Total Points |
|:---:|:---:|:---:|:---:|
| (G) | Gas Mileage | 50 | =50 |
| (A) | Attractiveness | 50 | =100 |
| (S) | Safety | 100 | =150 |
| (R) | Reliability | 250 | =300 |
| (P) | Purchase Price | 350 | =400 |

Step 4: Calculate Weights

Using the point scores assigned to each of the attributes in Step 3, the final step of the SMART process is to calculate attribute weights. This is done by normalizing each attribute score against the total score among all attributes as shown in Equation (3). In the car buying example, the total points distributed among all five preference attributes are 50 + 100 + 150 + 300 + 400 = 1000 points. The corresponding weights for each attribute are calculated as shown in Table 5. Note that this method can generate precisely the same weights as the DAT method (assuming the correct points are used in both).

**Table 5.** Weight calculation for SMART.

| Abbreviation | Criteria | Formula | Weight |
|:---:|:---:|:---:|:---:|
| (G) | Gas Mileage | 50/1000 | =0.050 |
| (A) | Attractiveness | 100/1000 | =0.100 |
| (S) | Safety | 150/1000 | =0.150 |
| (R) | Reliability | 300/1000 | =0.300 |
| (P) | Purchase Price | 400/1000 | =0.400 |
| | Sum | 1000 points | =1.00 |

Strengths of This Approach

SMART does not need to be repeated if old attributes are removed or new attributes are added, unless the one being removed is also the one that is least important or the one being added assumes the role of being the least important. The number of questions needed to assign weights using the SMART technique is equal to one less than the number of preference attributes. Thus, the effort required to obtain attribute weights scales linearly with the number of attributes.

Limitations of This Approach

The choice of the score for the lowest- (or highest-) weighted attribute may affect the resulting attribute weights if the scores for other attributes are not chosen based on relative comparisons. For example, if the least important attribute is given a value of 10 and some other attribute is given a value of 30, this latter value should increase to 60 if the baseline score given to the least important attribute is raised to 20.

3.1.3. Swing Weighting Techniques (SWING)

The Swing Weighting Technique [72] is an approach for determining weighting factors indirectly through systematic comparison of attributes against the one deemed to be the most important. SWING consists of two general activities: (1) rank order attributes according to the relative importance of incremental changes in attribute values considering the full range of possibilities; (2) select either the least or most important attribute as a reference point and assess how much more or less important the other attributes are with respect to the reference point. This step involves the calculation of attribute weights as the ratio of points assigned to an attribute to the total points assigned to all attributes.

The SWING Weighting technique is comprised of the following four steps: (1) rank order attributes, (2) establish the reference attribute, (3) estimate importance of other attributes with respect to the reference attribute, and (4) calculate weights.

SWING Step 1: Rank Order Attributes

Just as was the case with the SMART method, this method begins by rank ordering relevant attributes. For illustration purposes, we will use the same rank ordering as before (shown in Table 3).

SWING Step 2: Establish the Reference Attribute

In this second step, experts select a common reference attribute, assign it a fixed score, and estimate the extent to which the remaining attributes are more or less important than the reference attribute. Any attribute can assume the role of reference attribute. For the Swing Weighting method, however, this role is assigned to the most important attribute with a reference score of 100 points.

SWING Step 3: Score Attributes Relative to the Reference Attribute

Given a fixed reference attribute, experts are asked to estimate how much less important the remaining attributes are with respect to the reference attribute. For example, if the most important attribute is used as the reference point with a reference score of 100 points, experts would be asked to judge how many points should be allocated to each remaining

attribute with respect to this reference attribute in a relative sense (e.g., 10 less points) or an absolute sense (e.g., 90 points). It is common to systematically evaluate the remaining attributes in order of increasing or decreasing importance to ensure individual or group consistency between the results from this step and ordinal rankings from step 1; however, it may be worthwhile to randomize the order in which attributes are assessed as a means for uncovering any inconsistencies in preference.

Consider the car buying example discussed in Step 1. Using the most important attribute (purchase price) as the reference attribute, with a reference score of 100 points, point scores could be assigned to the remaining attributes as shown in Table 6.

**Table 6.** Ordinal ranking for SWING.

| Abbreviation | Criteria | Ordinal Ranking |
|---|---|---|
| (P) | Purchase Price | 100 |
| (R) | Reliability | 75 |
| (S) | Safety | 37.5 |
| (A) | Attractiveness | 25 |
| (G) | Gas Mileage | 12.5 |

SWING Step 4: Calculate Weights

Using the point scores assigned to each of the attributes in Elicitation Step 2, the final step of the SWING process is to calculate attribute weights. This is done by normalizing each attribute score against the total score among all attributes as shown in Equation (3). In the car buying example, the total points distributed among all five preference attributes are 12.5 + 25 + 37.5 + 75 + 100 = 250 points. The corresponding weights for each attribute can then be readily calculated as shown in Table 7. Note that the same weights are produced in this case as when using the previous two methods.

**Table 7.** Weight calculation for SWING.

| Abbreviation | Criteria | Formula | Weight |
|---|---|---|---|
| (P) | Purchase Price | 100/250 | =0.400 |
| (R) | Reliability | 75/250 | =0.300 |
| (S) | Safety | 37.5/250 | =0.150 |
| (A) | Attractiveness | 25/250 | =0.100 |
| (G) | Gas Mileage | 12.5/250 | =0.050 |
| | Sum | 250 points | =1.00 |

Strengths of This Approach

SWING considers the utility over the full range of attributes. SWING need not be repeated if old attributes are removed or new attributes are added unless the one being removed is also the one that is most important, or the one being added assumes the role of being the most important.

The number of questions needed to assign weights using the SWING technique is equal to one less than the number of preference attributes. Thus, the effort required to obtain attribute weights scales linearly with the number of attributes.

Limitations of This Approach

In contrast to the SMART technique, SWING weighting assigns a score with respect to a fixed upper score assigned to the most important attribute. While scores can be specified using any non-negative number up to the reference point, in practice, the presentation of the method often restricts users to specifying scores in terms of integer values. Consequently, users are limited to only 101 possible scores for each attribute, while, for SMART, the number of possible scores is infinite (e.g., 10 or higher). This means that SMART offers a greater diversity in weighting factor combinations than SWING.

The choice of the score for the most important attribute may affect the resulting attribute weights if the scores for other attributes are not chosen based on relative comparisons. For example, if the most important attribute is assigned a 100 and some other attribute is given a 50, this latter value should decrease to 40 if the baseline score given to the least important attribute is lowered to 80.

### 3.1.4. Simple Pairwise Comparison

The simple pairwise comparison technique for eliciting weights systematically considers all pairs of attributes in terms of which is more important. For each pairwise comparison, a point is assigned to the attribute that is considered more important. In the end, attribute weights are determined as the ratio of points assigned to each attribute divided by the total number of points distributed across all attributes.

The simple pairwise comparison technique is comprised of the following two steps: (1) pairwise rank the attributes, and (2) calculate weights.

### Pairwise Step 1: Pairwise Rank the Attributes

Given a set of $N$ attributes, systematically compare pairs of attributes in terms of which one of the two is more important relative to small changes over its range. Of the pair, the one judged to be more important is assigned a point. The process is repeated until all $N * (N - 1)/2$ pairs are evaluated. For example, consider the automobile purchase problem. Using the five criteria, there are $5 * (5 - 1)/2 = 10$ pairs to evaluate. The output from this pairwise ranking step might yield the following results:

- Purchase Price vs. Attractiveness: Purchase Price Wins;
- Purchase Price vs. Reliability: Purchase Price Wins;
- Purchase Price vs. Gas Mileage: Purchase Price Wins;
- Purchase Price vs. Safety Rating: Purchase Price Wins;
- Attractiveness vs. Reliability: Reliability Wins;
- Attractiveness vs. Gas Mileage: Attractiveness Wins;
- Attractiveness vs. Safety Rating: Safety Wins;
- Reliability vs. Gas Mileage: Reliability Wins;
- Reliability vs. Safety Rating: Reliability Wins;
- Gas Mileage vs. Safety Rating: Safety Wins;

The point distribution obtained using these 10 comparisons is shown in Table 8.

**Table 8.** Point calculation for pairwise method.

| Abbreviation | Criteria | Points |
|:---:|:---:|:---:|
| (P) | Purchase Price | 4 points |
| (R) | Reliability | 3 points |
| (S) | Safety | 2 points |
| (A) | Attractiveness | 1 point |
| (G) | Gas Mileage | 0 points |

Note that the least important attribute in the above example has a score of zero points (as it won none of the pairwise comparisons). The resulting weight factor in this case will be zero unless some constant offset or systematic bias is applied to all scores. Such an offset or bias desensitizes the resulting weights of the attributes to changes in the points distributed to each via a pairwise ranking procedure—the greater the offset, the less sensitive the resulting weighting distribution will be to small changes in attribute scores. For example, if an offset of 2 points or 10 points is used, the revised score distributions shown in Table 9 would result.

**Table 9.** Point calculation for pairwise method using offsets.

| Abbreviation | Criteria | Points (2/10 Offset) |
|---|---|---|
| (P) | Purchase Price | 6 points/14 points |
| (R) | Reliability | 5 points/13 points |
| (S) | Safety | 4 points/12 points |
| (A) | Attractiveness | 3 point/11 points |
| (G) | Gas Mileage | 2 points/10 points |

Pairwise Step 2: Calculate Weights

Using the point scores assigned to each of the attributes in the previous step, the second step of the simple pairwise comparison technique is to calculate attribute weights. This is done by normalizing each attribute score against the total score among all attributes using Equation (3). In our car buying example, the total points distributed among all five preference attributes is $N * (N − 1)/2$, or $5 * (5 − 1)/2 = 10$ points. The corresponding weights for each attribute can then be readily calculated as shown in Table 10. It should be noted that this method generates unique weights as compared with the previous three methods. This is due to the fact that the number of potential weights is more discrete (it is limited by the number of comparisons that are made) as compared to the previous methods.

**Table 10.** Weighs for pairwise method.

| Abbreviation | Criteria | Formula | Weight |
|---|---|---|---|
| (P) | Purchase Price | 4/10 | =0.4 |
| (R) | Reliability | 3/10 | =0.3 |
| (S) | Safety | 2/10 | =0.2 |
| (A) | Attractiveness | 1/10 | =0.1 |
| (G) | Gas Mileage | 0/10 | =0.0 |
| | Sum | 10 points | =1.00 |

To demonstrate the impact of imposing an offset or systematic bias to the attribute scores, the weights obtained from adding 2 points and 10 points to each are shown in Table 11.

**Table 11.** Weighs for pairwise method using offsets.

| Abbreviation | Criteria | Formula | Weight |
|---|---|---|---|
| (P) | Purchase Price | 6 points/14 points | =0.30/0.233 |
| (R) | Reliability | 5 points/13 points | =0.25/0.217 |
| (S) | Safety | 4 points/12 points | =0.20/0.20 |
| (A) | Attractiveness | 3 point/11 points | =0.15/0.183 |
| (G) | Gas Mileage | 2 points/10 points | =0.10/0.167 |
| | Sum | 20 points/60 points | =1.00 |

As the size of the offset or bias increases, the weights become equally distributed across attributes. In the case of an infinite offset, the approach results mirror the equal weighting technique.

Strengths of This Approach

This approach is very easy to complete since it requires users to judge which of two options is preferred. Such comparisons are much easier for humans to perform than ranking a complete list of attributes or assigning scores to each attribute. This approach also facilitates documentation of the reasoning supporting the resulting weight factors. It breaks down the questioning process to simple comparisons of two attributes that only requires evidence to support which attribute is more or less important than the other. The elicitation method requires that the user consider attribute ranges when making pairwise

judgments. Weights can be readily recalculated with the addition of new attributes simply by incorporating all additional pairwise comparisons.

Limitations of This Approach

This approach does not employ any checks of internal consistency (i.e., for transitivity). It is up to the user to check to see whether the results make sense and are consistent. For instance, if A > B (i.e., A is preferred to B), and B > C, then logically A > C; however, there is nothing in the process that ensures that intransitive assessments must be made.

A Special Case of Pairwise Comparison: The Analytic Hierarchy Process

The Analytic Hierarchy Process (AHP) is a common process used to elicit decision maker priorities using a series of pairwise comparisons [74]. Its use has increased in popularity [75] because of the ease of explanation, ease of facilitating a group in the process, and availability of user-friendly software to implement the process. However, it is generally not looked upon favorably within the decision analysis community because of several drawbacks. Velasquez and Hester [76] identify problems due to interdependence between criteria and alternatives, the potential for inconsistency between judgment and ranking criteria, and the possibility of rank reversal [77] as disadvantages of the method. It is worth noting that, with every weighting approach, there will be drawbacks.

### *3.2. Approximate Techniques*

Approximate Techniques establish weights based primarily on the ordinal rankings of attributes based on relative importance. Approximate techniques adopt the perspective that the actual attribute weights are samples from a population of possible weights, and the distribution of weight may be thought of as a random error component to a true weight [78]. As a result, approximate techniques seek the expected value of attribute weights and use these expected weights in utility models. A variety of approximate techniques exist, including: (1) equal weighting, (2) rank ordered centroid technique, (3) rank summed weighting technique, and (4) rank reciprocal technique.

### 3.2.1. Equal Weighting Technique

The Equal Weighting Technique assumes that no information is known about the relative importance of preference attributes or that the information pertinent to discriminating among attributes based on preference is unreliable. Under these conditions, one can adopt maximum entropy arguments and assume that the distribution of true weights follows a uniform distribution [79].

Given a set of N preference attributes, the Equal Weighting Technique assigns a weight $w_i$ to each attribute as shown in Equation (4).

$$w_i = 1/N \tag{4}$$

For example, consider our automobile purchase decision. Assuming no additional information is available to establish a preference ordering of the five problem attributes, an equal weight of 1/5 (0.20) is assigned to each.

Strengths of This Approach

The Equal Weighting Technique is the simplest of all weighting techniques, which includes both ratio assignment techniques and approximate techniques. The only prerequisite for applying the Equal Weighting Technique is a judgment that an attribute matters or is significant [78]. The Equal Weighting Technique is a formal name for what is naturally done in the early stages of analysis.

Limitations of This Approach

The weights resulting from application of the Equal Weighting Technique may produce inaccurate rankings if the true weights associated with one or more criteria dominate the others. As with any technique based on mathematical principles, the weights obtained

via the Equal Weighting Technique are only as good as its assumptions. The principle underlying the Equal Weighting Technique is the use of the uniform distribution constructed across all attributes. Alternative techniques should be used if this assumption is not applicable, or if more information exists that could assist in establishing a quantitative difference between attributes.

When some information is available to help distinguish between attributes on the basis of importance, alternative techniques will produce better estimates of attribute weights. When the number of attributes is 10 or less, it is more useful to spend resources to first establish a rank ordering of the attributes using group discussion or pairwise ranking and then follow-up with an alternative approximate techniques.

### 3.2.2. Rank Ordered Centroid (ROC) Technique

The Rank Ordered Centroid Technique assumes knowledge of the ordinal ranking of preference attributes with no other supporting quantitative information on how much more important one attribute is relative to the others [80]. As a consequence of this assumption, it is assumed that the weights are uniformly distributed on the simplex of rank ordered weights [78].

The Rank Ordered Centroid Technique is comprised of the following two steps: (1) rank order attributes and establish rank indices, and (2) calculate the rank ordered centroid for each attribute.

### ROC Step 1: Rank Order Attributes and Establish Rank Indices

Consider a finite set of N attributes or criteria deemed relevant by an individual or group of experts to a particular decision problem. This first step asks users to agree on a rank ordering of these attributes according to their relative contribution to the expert's overall preference within an additive utility (or value) function framework. A number of approaches exist to assist in holistic ranking, the most popular being pairwise ranking [81]. The resultant ranking is from most important to least important, where the index $i = 1$ is assigned to the most important attribute, and the index $i = N$ is assigned to the least important attribute.

For example, consider the typical choice problem centered on which automobile to purchase. The output from this step would be a rank ordering of these preference attributes from most to least important, as shown in Table 12.

**Table 12.** Ordinal criteria ranking.

| Abbreviation | Criteria | Ordinal Ranking with Index |
|:---:|:---:|:---:|
| (P) | Purchase Price | $i = 1$ |
| (R) | Reliability | $i = 2$ |
| (S) | Safety | $i = 3$ |
| (A) | Attractiveness | $i = 4$ |
| (G) | Gas Mileage | $i = 5$ |

### ROC Step 2: Calculate the Rank Ordered Centroid for Each Attribute

The Rank Ordered Centroid Technique assigns to each of $N$ rank ordered attributes a weight $w_i$ according to Equation (5).

$$wi = (1/N) \sum_{k=i}^{N} \frac{1}{K} \tag{5}$$

where, again, the attributes are ordered from most important ($i = 1$) to least important ($i = N$).

In our car buying example, the weights assigned to each attribute can be calculated as shown in Table 13. Note that the predefined formula used in the approximate techniques limits the number of weights available for assignment. Thus, while we can see the ordinality

of weight preferences remains, the magnitude of weights and distance between them has changed. This is the tradeoff that a decision maker must make; is more control over weights preferred or not? If so, using a ratio assignment technique provides more control. If time is more crucial or if decision makers are not as informed regarding the problem, an approximate technique may prove more appropriate.

**Table 13.** Weight using ROC technique.

| Abbreviation | Criteria | Formula | Weight |
| --- | --- | --- | --- |
| (P) | Purchase Price | $w_1 = 1/5 \, (1 + 1/2 + 1/3 + 1/4 + 1/5)$ | =0.457 |
| (R) | Reliability | $w_2 = 1/5 \, (1/2 + 1/3 + 1/4 + 1/5)$ | =0.257 |
| (S) | Safety | $w_3 = 1/5 \, (1/3 + 1/4 + 1/5)$ | =0.157 |
| (A) | Attractiveness | $w_4 = 1/5 \, (1/4 + 1/5)$ | =0.090 |
| (G) | Gas Mileage | $w_5 = 1/5 \, (1/5)$ | =0.040 |
| | Sum | $w_1 + w_2 + w_3 + w_4 + w_5$ | ~1.00 |

Strengths of This Approach

The Rank Ordered Centroid technique provides a means for coming up with meaningful weights based solely on ordinal rankings of attributes based on importance. This is particularly helpful since, in situations consisting of many users with diverse opinions, rank orderings of attributes may be the only aspect of preference that can achieve consensus agreement. Calculating weights using the Rank Ordered Centroid technique can be easily implemented using standard spreadsheet tools or calculated using a calculator.

Limitations of This Approach

When some information is available to help distinguish between attributes on the basis of importance, alternative techniques will produce better estimates of attribute weights. When the number of attributes is 10 or less, it is more useful to spend resources to first establish a rank ordering of the attributes using group discussion or pairwise ranking, and then follow-up with an alternative approximate techniques. As with any technique based on mathematical principles, the weights obtained via the Rank Ordered Centroid technique are only as good as its assumptions. The principle underlying the ROC technique is the use of the uniform distribution (justified by Laplace's principle of insufficient reason) across the range of possible weights that can be assumed by an attribute based on its importance rank. Alternative techniques should be used if this assumption is not applicable, or if more information exists that could assist in establishing a quantitative difference between attributes.

3.2.3. Rank Summed Weighting (RS) Technique

To approximate attribute weights, the Rank Summed Weighting technique uses information on the rank order of attributes on the basis of importance combined with the weighting of each attribute in relation to its rank order [61].

The Rank Summed Weighting technique is comprised of the following two steps: (1) rank order attributes and establish rank indices, and (2) calculate the rank summed weight for each attribute.

RS Step 1: Rank Order Attributes and Establish Rank Indices

Consider a finite set of N attributes or criteria deemed relevant by an individual or group of users to a particular decision problem. This first step asks users to agree on a rank ordering of these attributes according to their relative contribution to the expert's overall preference within an additive utility (or value) function framework. A number of approaches exist to assist in holistic ranking, the most popular being pairwise ranking [81]. The resultant ranking is from most important to least important, where the index $i = 1$ is assigned to the most important attribute and the index $i = N$ is assigned to the least

important attribute. For our car purchase example, we maintain the same ordinal ranking as shown in Table 12 for the previous method.

RS Step 2: Calculate the Rank Summed Weight for Each Attribute

The Rank Summed Weighting technique assigns, to each of $N$ rank ordered attributes, a weight $wi$ according to Equation (6).

$$wi = (N - i + 1)/\left(\sum_{k=1}^{N} N - i + 1\right) = 2(N - i + 1)/N(N + 1) \tag{6}$$

with the attributes ordered from most important ($i = 1$) to least important ($i = N$).

The rank exponent weighting technique is a generalization of the rank sum weighting technique, as shown in Equation (7).

$$wi = (N - i + 1)^p/\left(\sum_{k=1}^{N} N - i + 1\right)^p \tag{7}$$

In this case, a $p$ of 0 results in equal weights, $p = 1$ is the rank sum, and increasing $p$ values further disperses the weight distribution among attributes.

In the car buying example above, the weights assigned to each attribute can be calculated using the Rank Summed Weighting technique as shown in Table 14. Once again, ordinality of criteria preference remains when compared with previous methods; however, the spread of weights changes due to the predetermined rank summed formula.

**Table 14.** Weight using RS technique.

| Abbreviation | Criteria | Formula | Weight |
|:---:|:---:|:---:|:---:|
| (P) | Purchase Price | $w_1 = (2\ (5 + 1 - 1))/(5\ (5 + 1))$ | =0.333 |
| (R) | Reliability | $w_2 = (2\ (5 + 1 - 2))/(5\ (5 + 1))$ | =0.267 |
| (S) | Safety | $w_3 = (2\ (5 + 1 - 3))/(5\ (5 + 1))$ | =0.200 |
| (A) | Attractiveness | $w_4 = (2\ (5 + 1 - 4))/(5\ (5 + 1))$ | =0.133 |
| (G) | Gas Mileage | $w_5 = (2\ (5 + 1 - 5))/(5\ (5 + 1))$ | =0.067 |
| | Sum | $w_1 + w_2 + w_3 + w_4 + w_5$ | =1.00 |

Strengths of This Approach

The Rank Summed Weighting technique provides a means for coming up with meaningful weights based solely on ordinal rankings of attributes based on importance. This is particularly helpful since, in situations consisting of many users with diverse opinions, rank orderings of attributes may be the only aspect of preference that can achieve consensus agreement. Calculating weights using the Rank Summed Weighting technique can be easily implemented using standard spreadsheet tools or calculated using a calculator.

Limitations of This Approach

When some information is available to help distinguish between attributes on the basis of importance, alternative techniques will produce better estimates of attribute weights. When the number of attributes is 10 or less, it is more useful to spend resources to first establish a rank ordering of the attributes using group discussion or pairwise ranking, and then follow-up with alternative approximate techniques.

As with any technique based on mathematical principles, the weights obtained via the Rank Summed Weighting technique are only as good as their assumptions. The principle underlying the RS technique is the weighting of each attribute in proportion to its rank order in terms of importance. Alternative techniques should be used if this assumption is not applicable or unreasonable, or if more information exists that could assist in establishing a quantitative difference between attributes.

### 3.2.4. Rank Reciprocal Weighting (RR) Technique

The rank reciprocal method is similar to the ROC and RS methods. It involves use of the reciprocal of ranks, divided by the sum of the reciprocals [61].

The Rank Summed Weighting technique is comprised of the following two steps: (1) rank order attributes and establish rank indices, and (2) calculate the rank reciprocal weight for each attribute.

RR Step 1: Rank Order Attributes and Establish Rank Indices

Consider a finite set of N attributes or criteria deemed relevant by an individual or group of users to a particular decision problem. This first step asks users to agree on a rank ordering of these attributes according to their relative contribution to the expert's overall preference within an additive utility (or value) function framework. A number of approaches exist to assist in holistic ranking, the most popular being pairwise ranking [81]. The resultant ranking is from most important to least important, where the index $i = 1$ is assigned to the most important attribute, and the index $i = N$ is assigned to the least important attribute. For our car purchase example, we maintain the same ordinal ranking as shown in Table 12 for the previous method.

RR Step 2: Calculate the Rank Summed Weight for Each Attribute

The Rank Reciprocal Weighting technique assigns to each of the $N$ rank ordered attributes a weight $w_i$ according to Equation (8).

$$wi = (1/i) / \left( \sum_{k=1}^{N} 1/k \right) \tag{8}$$

where, again, the attributes are ordered from most important ($i = 1$) to least important ($i = N$). In the car buying example above, the weights assigned to each attribute can be calculated using the RR technique as shown in Table 15. Again, ordinality of criteria preference remains when compared with previous methods; however, the spread of weights changes due to the predetermined rank reciprocal formula.

**Table 15.** Weight using RR technique.

| Abbreviation | Criteria | Formula | Weight |
|:---:|:---:|:---:|:---:|
| (P) | Purchase Price | $w_1 = 1/(i \times \sum_{k=1}^{N} 1/k) =$ <br> $1/(1 \times ((1/1)+(1/2)+(1/3)+(1/4)+(1/5)))$ | =0.438 |
| (R) | Reliability | $w_2 = 1/(i \times \sum_{k=1}^{N} 1/k) =$ <br> $1/(2 \times ((1/1)+(1/2)+(1/3)+(1/4)+(1/5)))$ | =0.218 |
| (S) | Safety | $w_3 = 1/(i \times \sum_{k=1}^{N} 1/k) =$ <br> $1/(3 \times ((1/1)+(1/2)+(1/3)+(1/4)+(1/5)))$ | =0.146 |
| (A) | Attractiveness | $w_4 = 1/(i \times \sum_{k=1}^{N} 1/k) =$ <br> $1/(4 \times ((1/1)+(1/2)+(1/3)+(1/4)+(1/5)))$ | =0.109 |
| (G) | Gas Mileage | $w_5 = 1/(i \times \sum_{k=1}^{N} 1/k) =$ <br> $1/(5 \times ((1/1)+(1/2)+(1/3)+(1/4)+(1/5)))$ | =0.088 |
| | Sum | $w_1 + w_2 + w_3 + w_4 + w_5$ | ~1.00 |

Strengths of This Approach

Similarly to the rank ordered centroid and rank summed weighting approaches, the rank reciprocal technique provides a mechanism for calculating weights using only an ordinal ranking of relevant attributes. This technique is easily implemented using spreadsheet tools or calculated using a calculator.

Limitations of This Approach

As with the previous two methods, rank reciprocal weighting is best used when only an ordering of attributes is possible. When more specific weighting is possible, use of a ratio assignment technique is advised.

## 4. Discussion

Eight major techniques spanning the past several decades for computing weights in MCDA environments have been discussed. In a comparison of the categories of ratio assignment and approximate techniques, Jia, Fischer [78] found that the selection accuracy of quantitatively stated ratio weights was as good as or better than that of the best approximate methods under all conditions studied (except when the assessed weights are purely random). Because linear decision models are quite robust with respect to change of weights [40], using approximate weights yields satisfactory quality under a wide variety of circumstances. Despite the robustness of linear models, even noisy information about the ranking of attributes improves decisions substantially. When response error is present, decision quality decreases as the number of attributes or the number of alternatives rated against these attributes increases.

### 4.1. Characteristics of Multi-Criteria Decision Analysis Techniques

Knowing multiple ways to represent weightings with respect to making decisions provides numerous benefits for model building. It can help in evaluating, identifying, and selecting the best decisions for a given situation, whether this is provided as the primary model output or occurs frequently throughout execution as a component of the model's behaviors. Identifying and understanding different mechanisms for assigning weights helps to convey the complexities that can arise in modeling decision processes. This can be paired with verification and validation activities to provide a transparent connection between model design and simulation outcomes to aid in traceability and reproducibility [82,83]. Understanding the uses of the individual techniques can aid in the use of techniques based on the characteristics of the decision making within a modeled system.

Models can incorporate a combination of ratio assignment and approximate techniques and select the most appropriate method based on a given decision (refer to fourth column of Tables 16 and 17). Determining which criteria are important can help flush out a model's conceptualization and serve as supporting documentation for how and why certain variables are included in the model design. The observed advantages, disadvantages, and potential uses of each technique are summarized in Table 16 for ratio assignment techniques and in Table 17 for approximate techniques. How these techniques can inform the development of computational models is explored in Section 4.2.

### 4.2. MCDA as Decision-Making Options for Computational Models

Capturing and representing decision making processes is a common facet when constructing simulation models. Decision making can exist at many levels within a model, such as representing how an individual decides when to purchase a car, assisting a store manager in developing a personnel schedule for improved cost management, or for examining investment decisions. Simulations allow for observations on the performance of modeled behaviors to be conducted and analyzed [84] so that the modelers or decision makers can gain insight into whether the selected decision making processes led to the expected outcomes and to help them in making decisions based on these results. However, models that incorporate human decisions may produce unsuspected chaos as a result of a minority of the decision makers [85] and it can be challenging to identify what to capture and how to incorporate it within a model.

**Table 16.** Characteristics of weighting methods for ratio assignment techniques.

| Method | Advantages | Disadvantages | Uses |
|---|---|---|---|
| Direct assignment technique | Straightforward<br>Effort scales linearly with the number of attributes<br>Easily implemented with spreadsheet or calculator | Must be repeated if attributes change<br>Sensitive to reference point | Situations in which attributes have clear separation in terms of importance |
| Simple multi attribute rating technique (SMART)/SMARTER/SMARTS | Attributes can change without redoing assessment<br><br>Effort scales linearly with number of attributes<br>Greater weight diversity than SWING | Attribute value ranges influence weights | Situations in which attributes have clear separation in terms of importance<br>Scenarios where scales for attributes are clear |
| Swing weighting | Attributes can change without redoing assessment<br><br>Effort scales linearly with number of attributes | Limited number of weights available | Situations in which attributes have clear separation in terms of importance<br>Scenarios where scales for attributes are clear |
| Simple pairwise comparison | Low effort | Does not prevent weight inconsistency | Situations in which attributes have clear separation in terms of importance<br>Scenarios where scales for attributes are clear |

**Table 17.** Characteristics of weighting methods for approximate techniques.

| Method | Advantages | Disadvantages | Uses |
|---|---|---|---|
| Equal weighting | Easiest of all methods<br><br>Easily implemented with spreadsheet or calculator | Few if any real world scenarios have all attributes of equal importance<br>Inaccurate relative to other techniques | Early in the decision process<br><br>Situations with incomplete or no attribute information<br>Scenarios where a large number of attributes are present |
| Rank Ordered Centroid | Uses ordinal ranking only to determine weights<br><br>Easily implemented with spreadsheet or calculator | Based on uniform distribution | Analyst is unwilling to assign specifics weights<br>Scenarios when consensus may not be necessary or desirable, but ranking can be agreed upon [80]<br>Scenarios where a large number of attributes are present |
| Rank Sum | Uses ordinal ranking only to determine weights<br><br>Easily implemented with spreadsheet or calculator | Based on uniform distribution | Analyst is unwilling to assign specifics weights<br>Scenarios when consensus may not be necessary or desirable, but ranking can be agreed upon [80]<br>Scenarios where a large number of attributes are present |
| Rank Reciprocal | Uses ordinal ranking only to determine weights<br><br>Easily implemented with spreadsheet or calculator | Only useful when more precise weighting is not available | Analyst is unwilling to assign specific weights<br>Scenarios when consensus may not be necessary or desirable, but ranking can be agreed upon [80]<br>Scenarios where a large number of attributes are present |

The applicability of the ratio assignment and approximate techniques differ based on the context of the problem being addressed, the decisions being made, the decision makers being modeled, and the criterion that have been deemed necessary for a given decision. We provide an overview of the components within ABM, SD, and DES that are relevant to implementing these techniques within a simulation model. Table 18 provides a comparison of ratio assignment techniques and Table 19 provides a comparison of approximate techniques. These tables are intended to provide guidance and initial steps towards incorporating MCDA techniques. These are not intended to be exhaustive comparisons.

**Table 18.** Considerations for incorporating ratio assignment techniques into ABM, DES, and SD modeling paradigms.

| Ratio Assignment Technique | Agent Based Modeling | Discrete Event Simulation | System Dynamics |
|---|---|---|---|
| Direct assignment technique | Known * or accepted ˆ criteria that direct an agent towards their goals or one decision outcome or another | Known or accepted decision path probabilities; Known or accepted resource schedules | Known or accepted coefficient values within ordinary differential equation (ODE), partial differential equation (PDE) or difference equation (DE) |
| Simple multi attribute rating technique (SMART)/ SMARTER/ SMARTS | There exists an accepted least important criterion and the remaining criteria are weighted relative to this option. Each agent population may utilize difference weighting preferences. | A least acceptable path is known and the remaining options are weighted relative to this option. Weighting preferences can vary by entity type. | The ODE, PDE, or DE contains a value whose coefficient is known to be least important. Remaining coefficients are weighted relative to this coefficient. |
| Swing weighting | Order of importance is known/accepted but the most important element is not always the top ranked. Current rankings and known important criterion are used to establish weightings of remaining criteria. | Top ranked path or most desirable schedule are known but do not always remain top ranked during execution. Selections are made relative to the known choice based on its current ranking. | Coefficient weightings are intended to weight towards a specified most important criterion; however, new weights are generated based on magnitude of change from previous check to incorporate stochasticity. |
| Simple pairwise comparison | No established known or accepted ranking of criteria weightings. Agent compares all available criteria to accumulate weighting scores. | No established known or accepted ranking of criteria weightings. Entities or resources compare all available criteria to accumulate weighting scores for path probabilities or scheduling. | No established known or accepted ranking of criteria (e.g., coefficient) weightings. Equation coefficient weightings accumulate based on comparisons of all criteria. |

* The term *known* reflects that a weighting is supported by empirical evidence. ˆ The term *accepted* reflects general agreement among the model's builders and stakeholders.

Determining the appropriate MCDA technique to select is highly dependent upon the given system context, the outcomes being examined, any performance metrics being assessed, the level of aggregation desired, and many other potential criteria. The criteria that are involved in the weighting combinations serve as candidates that should be involved in the verification and validation stages of model development and testing. For verification, the implementation of the criteria should be traceable back to the MCDA technique identified in the model design. This should be checked for consistency against the subject matter experts' specifications or any other conceptual model documentation. For validation, the selected MCDA categorization as well as the specific technique and its method for distributing weights, should be checked against what is known about the system. The determination of whether the decision process from the real system is more accurately represented as a ratio assignment or as an approximation should be defendable based on the data known about the system. This will reinforce the credibility of the technique selection and the model construction.

**Table 19.** Considerations for incorporating Approximate Techniques into ABM, DES, and SD modeling paradigms.

| Approximate Technique | Agent Based Modeling | Discrete Event Simulation | System Dynamics |
|---|---|---|---|
| Equal Weighting | Agent decision criterion is assumed of equal importance. This technique may be applicable in cases where the use of the uniform distribution for sampling is appropriate. | Path selection or resource selection is assumed of equal importance. This technique may be applicable in cases where the use of the uniform distribution for sampling is appropriate. | Values of coefficient weightings are assumed of equal importance. |
| Rank Ordered Centroid Technique | Order of importance of decision criterion are based on the aggregate orderings from each agent and update over time. | Resource schedules depend on aggregate rankings of criterion from the entities or resources which change as resource availabilities (e.g., through schedules) change or as aggregated weight and processing times change. | Values of coefficient weightings are based on the aggregate performance of stock or auxiliary variable performance over time. |
| Rank Sum Technique | Weightings are based on aggregated rankings of importance from each agent based on a utility function. | Weightings are based on aggregated rankings of importance from each entity over time based on a utility function. | Weightings are based on aggregated rankings of importance of stocks or auxiliary variables over time based on a utility function. |
| Rank Reciprocal | Weightings are based on aggregated rankings of importance from each agent based on preference. | Weightings are based on aggregated rankings of preferred importance from each entity per entity type. | Weightings are based on aggregated rankings of importance of stocks or auxiliary variables over time based on preference. |

Many simulation platforms natively allow for some instance of an equal or percent-based choice to be implemented within the model, whether these choice options exist at the system level in the form of flows [86], at the process level in the form of path logic [26,87], or at the individual level to capture the decisions of individual agents [43]. However, the implementation of decisions that are based on the rankings, weightings, or comparisons between multiple attributes is not generally as straightforward of a task. Tables 18 and 19 are intended to inform the model builder of circumstances under which the reviewed approximate and ratio assignment techniques may be of use. Simulation and domain expertise are still required to properly implement and test the technique. The application of a MCDA technique within a simulation should fill a necessary gap, maintain traceability to the model's requirements, and not introduce new gaps or unnecessary challenges to the simulation [82,88–90].

How criteria weightings are conceptualized and how they are implemented in practice can vary greatly across modeling paradigms. For instance, consider the equal weighting method from the approximate techniques. While this may be a conceptually simple technique to conceptualize weighting assignments for a given set of criteria, the considerations for which criterion are important, how the criterion are interconnected, and the potential results that can come out of the decision may be very different. This can result from differences in desired levels of scale and aggregation, continuous or discrete representation of components, and the desired time advance granularity [88,89,91–94]. Sections 4.2.1–4.2.3 discuss the potential applicability of utilizing ratio assignment and approximate techniques within the ABM, DES, and SD modeling paradigms.

### 4.2.1. MCDA Applicability for Agent-Based Models

As weighting methods are based on what is deemed to be important by the decision maker, MCDA can provide several unique benefits to ABM. Agent populations are commonly heterogeneous, spatially separate from their environment, dynamic, and behave based on agent–agent and agent–environment rules [28,29,84]. Diverse methods, algo-

rithms, and selection criteria represent decision making opportunities in ways that are more representative of the system being modeled [43,95]. Simulated agents can perform large volumes of decisions throughout their lifetimes, and they are constantly seeking to meet their goals, follow through on behaviors, and progress through life states. Based on the results of their cumulative interactions, their current and past experiences, and by accomplishing and/or redefining their objectives over time, such as through achieving goals and forming new goals, agents' weighting criteria may change as well. For instance, an agent faced with a decision about allocating his or her funds may not have a clear ranking of importance while happy and sufficiently wealthy; however, that agent may have a well-established importance ordering while unhappy and lacking wealth [29]. The agents' internal logics can change the weighting criteria method being utilized to better reflect their current states over time to achieve a more realistic representation of the system.

Different specifications of agent behaviors can lead to similar outcomes, and agent-based models can assist in identifying which agent behaviors provide the most simple explanation of the system behavior [28]. The agent-based modeling paradigm provides the ability to observe system level behaviors that result from each individual agent making individualized decisions based on local knowledge and personal perspective. Decision categorizations can include combinations of emotional, cognitive, and social factors [96]; personality traits in the face of life threatening environmental stimuli [97]; communality and affinity for selecting group formations [98]. Recognition-primed decision making has been employed to represent the decision process of a senior military commander to reflect the variability of humans in making decisions within an operational military environment where problems are commonly complex and complete information is not often known [41]. Knoeri, Nikolic [99] construct a model using awareness and incentives to enhance decision processes for recycling materials to explore the effects on construction wastes. Balke and Gilbert (43) provide a comprehensive and comparative review of 14 architectures for agent decision making that focuses on the architectures' cognitive, affective, social, learning, and norm consideration features.

Incorporating MCDA techniques into decision processes can benefit the ways that social norms are represented within a population, increase fidelity based on the environmental weightings that pertain to individual decisions, and represent geographical factors as weighting mechanics within the context of personal decision factors. Norms represent the certain ways that people act within a society and how they are punished when they act differently [100]. Norms representations and their effects on the population vary from modeling cooperation among unrelated individuals [101], to anxiety between group affiliations [34], to environmental and social stressors [97]. The level of agreement among the model's builders and stakeholders, the availability of supporting empirical evidence, and the number of relevant decision attributes should be considered when evaluating applicable MCDA techniques. Due to the potentially large number of agents and decisions, the computational complexity of the weighting technique and how often weightings are recalculated should also be factored into the technique selection.

### 4.2.2. MCDA Applicability for Discrete Event Simulation Models

Discrete Event Simulation models generally represent decisions from an aggregate level where mechanisms for defining entity movement or routing are specified. The entities moving through the system have no control over the decision itself; instead, progression-based decisions are made for the entities at the system level using percentages or logical determinations, such as entity type, percent chance, or shortest queue lengths [24–26,102,103]. Starting at the conceptual model building phase, focusing on DES decision elements aids in the identification of relevant criterion, fuels learning and collaboration, and contributes to assessing model validation [25,104,105]. While the static structure of DES simulations generally dictates the paths that one can follow, many simulation platforms allow for the incorporation of logic within entities to allow greater depth in path selection [26,106].

MCDA can be utilized to determine the weights of the transition options coming out of decision nodes, for representing path-selection logic within entities, and for determining or altering resource schedules. Decision attributes that have clear separations of importance may be better represented using ratio assignment techniques. Attributes with assumed equal weightings or weightings based on sampling from the uniform distribution may be more suitable when using approximate techniques. The number of attributes present in the relevant decision-making processes and the number of entity types, as well as the level of agreement within the simulation's conceptual model, its simulation building team, or its stakeholders, should be evaluated within the context of the problem being modeled to select the appropriate technique within the corresponding MCDA technique category. Within the domain of healthcare, MCDA techniques can provide alternative means for modeling staff scheduling, patient admission, patient routing, and resource allocation.

A survey of simulation application priorities emphasizes the relevance of human performance modeling, modeling complex behavior, and human decision-making towards the health care and service industries [107]. Within the scope of quantitative methods, DES models can incorporate resources and constraints, include soft variables from surveys and expert opinions, and cope with the high levels of variabilities existing between and within variables [108]. Data such as patient arrival times, discharge times, bed types, and time to bed within an emergency department are common variables when examining system performance and exploring improvements [24]. This type of criteria could be utilized for constructing ratio assignments or approximate techniques within a simulation to drive simulated decisions.

### 4.2.3. MCDA Applicability for System Dynamics Models

Decision-making in SD models is generally represented through the flows that connect stocks and are implemented in the forms of ordinary differential or partial differential equations [22]. As such, a decision criterion is represented within an equation in the form of a variable, with its coefficient representing the weighting. These coefficients can be constants established at initialization or change dynamically throughout execution. In SD platforms, these are commonly represented as stocks and auxiliary variables. MCDA techniques can be incorporated to handle situations where dynamic weightings are needed based on aggregate states of the SD simulation, different possible interpretations of auxiliary variables, or as a result of structural changes to the simulation.

In SD models, decision environments are represented through the dynamic behavior of the system based on what is known about the state of the system using its variables and inventory levels [22]. These models have been used to identify, evaluate, and assist in making economic decisions for a variety of systems, such as for enhanced oil recovery operations [109] for industrial production and distribution systems [110], and for inventory logistics within healthcare systems [111]. The aggregate decision-making representations of these models can result in chaos due to the human decision-making behaviors of a significant minority within the model [85].

MCDA has been utilized within SD models of health preparedness for pandemic influenza to evaluate mitigation strategies based on epidemiological parameters and policy makers' prioritizations [112]. The integration of MCDA with SD has been successful in representing multiple goals, objectives, and perspectives for community-based forest management [113]. A review of sustainable supply chain management identifies that the integration of SD with MCDA can help to address the identified scientific rigor shortcoming of neglected model validation and the disclosure of model equations [114]. Selecting suitable MCDA techniques based on Tables 16 and 17 requires considering the quantity of variables, any consensus among stakeholders, and the availability of empirical data to inform validation.

*4.3. Limitations*

This article does not consider strategies for assessing weight factors in other choice frameworks (e.g., ordered–weighted averaging or multiplicative utility models), nor does it consider techniques for obtaining the coefficients of linear models, proper or improper, in general. This research also does not focus on defining value functions for use in Equation (1), but, rather, our focus is on how a decision maker should best determine the appropriate weights for different criteria in a MCDA problem.

## 5. Conclusions

When faced with many attributes, it is often more convenient to use approximate techniques for assigning attribute weights, in the absence of more information. One can also use approximate techniques for an initial weighting and further refine using a ratio assignment technique. Whenever possible, rationale should accompany any judgments leading to attribute weights. Rationale includes documenting the information and reasoning that support each individual judgment, even if it is based strictly on intuition. Providing justification increases model transparency and exposes the model to critical review.

Further, when possible, it is useful to apply more than one technique for eliciting weights of preference attributes. If the results following the honest application of two or more techniques are the same, the credibility of the corresponding utility model is increased. In contrast, if the results are not the same, the disagreement provides a basis for settling differences in opinion, discussing model limitations and assumptions, and diagnosing hidden biases.

The use of MCDA allows for the creation of more realistic or granular representations of decision-making processes for computational models. We have provided a classification of ratio assignment and approximate techniques for conducting MCDA along with an evaluation of the strengths and weaknesses of each technique. The characteristics supporting the suitability of a given ratio assignment or approximate technique under a given context and modeling paradigm are discussed. Model building considerations that should be accounted for in applying MCDA techniques within computational models in practice are presented for ABM, DES, and SD modeling paradigms.

Future work is needed to evaluate other categories of MCDA techniques and how they support model conceptualization, implementation, verification, validation, and analysis. Incorporating MCDA techniques into model decision processes aids in traceability between the developed simulation and the modeled systems. This can aid verification and validation practices in determining the correctness of the implemented simulation and accuracy of the representation of the real system. Additional research is needed into the connections between documenting MCDA development within a simulation and effective means for utilizing it to aid conceptualization, verification, and validation.

The ability to determine the criteria that should be involved in a decision, how to potentially approach weighting the criteria, and how to validate the weightings are dependent upon system knowledge, stakeholder knowledge, and empirical evidence. To this end, social media platform usage continues to increase the volume of easily accessible personal information being directly posted about peoples' daily activities, key events, and their likes and dislikes. As a result, there are growing possibilities for connecting simulations directly into the "human" component of data by utilizing these sources of real information for deriving decision-making criteria. Kavak, Vernon-Bido [115] explore the use of social media data in simulations as sources of input data, for calibration, for recognizing mobility patterns, and for identifying communication patterns. Padilla, Kavak [116] use tweets to identify individual-level tourist visit patterns and sentiment. Recent advances explore the characteristics comprising sentiment-based scores utilizing posted information on twitter [117] and through YouTube videos [118]. These information sources can provide new avenues towards identifying decision criteria and desired outcomes, and in developing individual-level and population-based behaviors and rules which can further fuel the use of MCDA within existing modeling paradigms.

Ultimately, there is no one universal "right" way to conduct weighting for a MCDA problem. As discussed earlier, ordinality is preserved when using any of the techniques correctly. However, more coarse weights can be determined using approximate techniques and more refined weights are possible using ratio techniques. Which method is appropriate depends on the problem context. This article benefits practitioners by providing a comprehensive review and comparison of common weighting methods that can help to guide the selection of weighting methods to better address the questions being asked of a modeled system.

**Author Contributions:** Conceptualization, B.E., C.J.L. and P.T.H.; methodology, B.E., C.J.L. and P.T.H.; writing—original draft preparation, B.E., C.J.L. and P.T.H.; writing—review and editing, B.E. and P.T.H. All authors have read and agreed to the published version of the manuscript.

**Funding:** This research received no external funding.

**Institutional Review Board Statement:** Not applicable.

**Informed Consent Statement:** Not applicable.

**Data Availability Statement:** Not applicable.

**Conflicts of Interest:** The authors declare no conflict of interest.

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
