# Peer review of "Methods for Weighting Decisions to Assist Modelers and Decision Analysts: A Review of Ratio Assignment and Approximate Techniques"

_applsci, doi:10.3390/app112110397_

Round 1

Reviewer 1 Report

This article reviews and summarizes eight multi-criteria decision analysis (MCDA) techniques that serve as options for reaching unique decisions based on personally and individually ranked criteria. In general, the article has been well written. In think it can be considered for publication after some necessary revisions and improvements. Some of my comments are below:

--The contribution of this article should be summarized in a clear way.

--Please clarify the criteria of the taxonomy of MCDA techniques.

--The literature review regarding MCDA is not comprehensive. Please improve it. I think the following papers may be helpful for fix this issue: Ranking range based approach to MADM under incomplete context and its application in venture investment evaluation. Technological and Economic Development of Economy 25 (2019) 877-899; Exploring the ordinal classifications of failure modes in the reliability management: An optimization-based consensus model with bounded confidences. Group Decision and Negotiation, in press (2021), DOI: 10.1007/s10726-021-09756-9; An overview on feedback mechanisms with minimum adjustment or cost in consensus reaching in group decision making: Research paradigms and challenges. Information Fusion 60 (2020) 65-79.

Reviewer 2 Report

Dear Editor,

Thank you authors for the interesting Manuscript ID applsci-1397200.with a title "Methods for Weighting Decisions to Assist Modelers and Decision Analysists: A Review of Ratio Assignment and Approximate Techniques". The manuscript presented with a good level of the presented research. The research introduction, research design and methods presented correctly. Several comments for the improvement are follow:

  1. I recommend the research present as review article;
  2. In the review of Ratio Assignment and Approximate Techniques must be included more most popular methods for example expert judgement and other methods;
  3. The comparison between selected methods with a proposal for the recommendation can by presented in the manuscript;
  4. The newly literature source (2022 year) be selected topic of the problem can be presented in the literature analysis and reference parts.

Reviewer
